# Perception of Concern and Associated Factors During the COVID-19 Pandemic: An Epidemiological Survey in a Brazilian Municipality

**DOI:** 10.3390/ijerph22081293

**Published:** 2025-08-19

**Authors:** Adriano Pires Barbosa, Marília Jesus Batista

**Affiliations:** Faculdade de Medicina de Jundiaí, 250, Francisco Teles St., Vila Arens II, Jundiaí 13202-550, Brazil; adrianobarbosa@g.fmj.br

**Keywords:** COVID-19, epidemiological monitoring, expression of concern, socioeconomic impact

## Abstract

The COVID-19 pandemic impacted mental health. This cross-sectional study analyzed the perception of concern related to the pandemic in Jundiaí-SP, June 2020. Participants consisted of residents of households selected by probability sampling and individuals with respiratory symptoms who sought Basic Health Units (UBS). The participants underwent rapid testing for SARS-CoV-2 and answered a questionnaire. The outcome was the perception of concern about pandemic and the independent variables were socioeconomic characteristics, behavioral variables, signs and symptoms, and rapid test results. Bivariate analysis was performed and variables with *p* < 0.20 were included in a binary logistic regression model (*p* < 0.05) using SPSS 20.0. A total of 2432 individuals participated in the study, including 1181 from UBS and 1251 from households. Females (OR: 1.42; CI: 1.18–1.71), black and mixed race participants (OR: 1.40; CI: 1.15–1.71), participants with an income up to 3 minimum wages (MW) (up to 1 MW: OR: 2.58; CI: 1.80–3.70; 1 to 3 MW: OR: 1.64; CI: 1.35–1.98), and younger participants (18–39 years: OR: 3.07; CI: 2.39–3.94; 40–59 years: OR: 2.42; CI: 1.89–3.10) were more concerned. Greater concern was perceived by more vulnerable individuals, regardless of testing positive for COVID-19 which is important to subsidize public mental health policies and crisis interventions, focusing on reducing race, gender and socioeconomic inequalities.

## 1. Introduction

Epidemics and pandemics, which are an important part of human history [1,2], cause changes in behavior and habits and introduce ways of coping with diseases, i.e., they produce psychosocial changes that need to be addressed. One such example is the latest COVID-19 pandemic, that affects human daily life and presented a great impact on mental health [Figaro]. The term ‘psychosocial’ describes the influence of social factors on an individual’s mental health and behavior [3].

It was observed that in addition to serious organic disease and mortality in Brazil [4], psychological changes are also part of the spectrum of problems caused by the new coronavirus and the pandemic. Studies conducted in China have shown that 52.1% of participants felt very afraid or apprehensive due to the pandemic and 53.8% classified the impact of the pandemic as moderate or severe; 16.5% reported moderate or severe depressive symptoms and 28.8% reported moderate or severe anxiety symptoms [5,6].

Such psychosocial changes caused by the pandemic were not only observed in China. The American Psychiatric Association (APA) published a survey, which showed psychosocial aspects related to pandemic, that almost half (48%) of Americans felt anxious about the possibility of contracting the new coronavirus and 40% were anxious about the possibility of developing serious illness or dying from the virus [7]. Furthermore, 62% of Americans were anxious about the possibility of a family member or loved one contracting the coronavirus. In a study conducted in Denmark [8] concluded that the pandemic had a negative impact on the psychosocial well-being of Danes and this impact was more frequent in women than in men.

The measures necessary to contain the virus themselves can be reasons for psychosocial changes. Isolation can cause depression and anxiety, especially for children; the closure of schools and increasement of stress and cases of violence [9]. Regarding the economic situation, the closure of companies caused concerns for business owners and employees, unemployment, financial crises, domestic violence, substance abuse, and social isolation [10]. The psychosocial changes of COVID-19 are therefore multifactorial and are observed in multiple layers [11] which means that beyond the health problems caused by the infection, the pandemic has broken out problems as stress, anxiety, depressive symptoms, insomnia, denial, anger and fear globally. According to this review [11] collective concerns influence daily behaviors, economy, prevention strategies and decision-making from policy makers, health organizations and medical centers, which can weaken strategies of COVID-19 control and lead to more morbidity and mental health needs at global level, with consequences even after the pandemic. Difficulties in adapting to pandemics and imposed quarantines can produce symptoms such as anxiety, fear, frustration, loneliness, anger, boredom, depression, stress, and avoidance behaviors [1].

To understand the psychosocial changes caused by a pandemic, it is important to observe the associated emotions such as fear, anger [12] and concern. A Brazilian study evaluated fear of COVID-19 and diverse psychological symptoms such as anxiety, depression, avoidance, physical symptoms, and functional loss. Among the 1844 individuals interviewed, 41% and 13.2% reported moderate or severe fear of COVID-19, respectively [13].

Although COVID-19 is a relatively recent disease, several authors have already demonstrated the effects of this pandemic on psychosocial aspects. Social isolation, anxiety, fear of contagion, feeling of uncertainty, and economic difficulties can lead to the development or exacerbation of depression, anxiety, substance abuse, or other psychiatric disorders in vulnerable populations, including individuals with pre-existing psychiatric illnesses and those living in regions with a high prevalence of COVID-19 [1,2,4,5,6,7,8,9,10,11,12,13]. The feeling of worry, that is the concern, may be a common denominator among all psychosocial changes mentioned.

Concern is described as a special state of the cognitive system whose function is to anticipate future danger [14]. It is a frequent cognitive activity that can range from “useful” concern to catastrophic, repetitive, and debilitating speculation [15]. Concern can be the expression of poor adaptation to a situation and is also a central symptom of anxiety disorders [16]. Few studies in Brazil have addressed concern during the pandemic as a factor that generates anxiety and stress. Particularly in Brazil, it is relevant to study which aspects increase the burden on the population. The socioeconomic aspects involved in the epidemic context can bring great impact because it is a country of great social inequalities [17] and part of the population is affected in many ways in situations like pandemics, environmental disasters, or crisis situations.

Understanding the relevance of these issues, in 2020, the World Health Organization (WHO) proposed to European member countries a tool designed to assist studies on behavioral changes caused by the pandemic in their populations [18]. This instrument, called Behavioural Insights on COVID-19, was translated and adapted to Brazilian Portuguese for application in a seroepidemiological population-based survey of COVID-19 in Jundiaí [19].

The aforementioned data can be used to indicate that concern, when dysfunctional, can lead to more serious psychosocial changes. However, few studies have investigated concern caused by the pandemic; addressing this situation is therefore necessary to better understand it and its psychosocial impacts. Therefore, the aim of this study was to analyze the impact of the COVID-19 pandemic on the perception of concern in the population of Jundiaí-SP and associated factors.

## 2. Methods

The present study is a branch of a major seroepidemiological population-based survey of COVID-19 conducted in Jundiaí that aimed to trace the epidemiological and seroepidemiological profile, as well as to investigate health behaviors during the pandemic and the population’s perception of concern. Details of this study can be found in Batista et al. [19].

The cited major survey was a cross-sectional study conducted based on the Strengthening the Reporting of Observational Studies in Epidemiology (STROBE) protocol for cross-sectional studies. The study was conducted in the municipality of Jundiaí, State of São Paulo, Brazil, between 1 and 20 June 2020. The municipality has a territory of 431,207 km^2^ and is divided into seven regions (center, east, northeast, northwest, north, west, and south). In 2021, Jundiaí had an estimated population of 426,935 inhabitants according to the Brazilian Institute of Geography and Statistics (IBGE). According to IBGE, the Municipal Human Development Index (MHDI) was 0.822 in 2010, which is considered to be very high compared to national standards. The municipality of Jundiaí is divided into four main health regions and has 35 Health Units, in addition to the Community Center and the Testing and Counseling Center [20].

Sample size calculation and sample selection were performed for the main outcome of testing positive for COVID-19, with two sampling stages to achieve the objective of the main study and to obtain representativeness of the population. All participants answered the same questionnaire and underwent the same rapid COVID-19 test. Interviewers were trained by the research team in order to perform the questionnaire assessment in a standardized manner.

For the stage carried out at the Basic Health Unit (UBS), the sample size was calculated considering a prevalence of positive tests of 10% among all individuals with respiratory symptoms tested, an alpha value of 5%, a power of the test of 95%, and an odds ratio (OR) of 3.0. The defined sample size was 940 individuals; 20% were added to compensate for possible losses, totaling 1175 cases. The sample of this stage was selected by including individuals attending the Unified Health System (SUS) because of symptoms suggestive of flu syndrome and who had a rapid test scheduled at 14 days after the onset of symptoms. Participants were selected by convenience sampling. A target number of participants was estimated for each UBS based on the weekly average number of rapid tests performed in the weeks prior to the start of this study. On the occasion of the rapid test, individuals were invited to participate in the study by a member of the team; once they had agreed to participate, the individuals underwent the rapid test and answered the questionnaire. The inclusion of individuals in the study and the application of the questionnaire were stopped when the UBS reached the established number of participants (Figure 1).

At UBS, individuals underwent rapid testing and were invited to answer a questionnaire which was developed based on the Behavioral Insights tool, recommended by the WHO [18].

In the household stage, sample size calculation considered a prevalence of 50%, an estimated population of 400,000, an alpha value of 5%, and a margin of error of 3%, totaling 1067 households. To compensate for possible sample losses, 20% were added and the final sample consisted of 1333 households (Figure 1) [19].

The participants were residents of the selected households who agreed to the rapid test and to answer the questionnaire. The households were selected using self-weighted probability sampling. First, a list of all neighborhoods and their respective number of households was obtained from the city hall, updated based on the Urban Land and Property Tax registration. This list was used for initial sample size calculation for each neighborhood considering all neighborhoods in the municipality, with the probability proportional to the number of households. Next, the households in each neighborhood composing the sample were randomly selected using Microsoft Excel^®^ according the address (Figure 1). Considering the possibility of refusals and/or empty homes, two replacements were randomly selected for each previously selected household head. During the home visit for rapid testing, one person from the household was invited to participate in the study and, after agreeing to participation, underwent the rapid test and answered the questionnaire.

The questionnaire was developed based on the Behavioral Insights recommended by WHO [18], adapted for the study with the addiction of sociodemographic questions.

The outcome of the study was the participant’s perception of concern caused by the COVID-19 pandemic, which comprise different aspects that contribute to an individual’s psychological well-being and that can be affected in times of crisis, including social and family life and financial aspects [18]. The original questionnaire has 25 pages suggested with topics in each page as sociodemographic questions, knowledge and others as worry in page 22. These topic start with the phrase “Crises often involve fears and worries”. Please let us know: “and then presented fourteen options”. We selected nine of these options that were relevant to our context, such as fear of being alone, fear of losing someone close due to the disease, fear of a family member or friend contracting coronavirus, fear of going through financial difficulties, school closure, fear of catching coronavirus, fear of becoming unemployed, fear of dying, and fear that the pandemic would cause a shortage of food and basic items.

The concern was assessed based on the degree of worry that the respondent exhibited in certain situations during the pandemic. This assessment consisted of nine questions whose answers were rated as “does not worry me” (1), “worries me a little” (2), and “worries me a lot” (3). Scores were assigned to the responses on a 3-point psychometric scale (Likert). The total score was obtained by summing the scores of these nine questions and was dichotomized according to the median obtained from the results of the study, which was 22. A score above the median was classified as “worries me a lot” and a score at the median or below as “does not worry me” or “worries me a little”.

The following independent sociodemographic variables of the respondents were analyzed: sex, self-reported as male or female; age in years, grouped into 3 categories; self-reported skin color; education as complete levels, and mean income as minimum wage (MW) at the time of the study. The following household variables were also analyzed: the household’s highest educational level, whether or not there was a health professional in the household, whether income was affected by the pandemic, and whether someone lost a job due to the disease.

In addition to the variables mentioned above, the following variables were analyzed by the Jundiaí seroepidemiological survey of COVID-19 [19]: presence or absence of comorbidities; regarding COVID-19: presence or absence of symptoms, the types of symptoms, and the date of onset; whether or not the participant had been previously tested, test date, and result; hospitalization; knowledge about the infection and its prevention; access to health information; adherence to social isolation, and loss of someone due to the disease.

The results were analyzed descriptively using SPSS^®^ 20.0 for characterization of the sample of the two stages and the total sample. The total data of the study were used for the analysis (data from households and Basic Health Units), with all participants answering the same questionnaire and being tested using the same method and during the same period. For the scale of the outcome (nine questions of concern) the Cronbach’s α were estimated, according to which the instrument may be considered moderate if its internal consistency between 0.60 and 0.75 [21]. Bivariate analyses between the outcome (Perception of concern) and the independent variables were then performed considering the COVID-19 test result, presence or absence of symptoms of the disease, sociodemographic and impact of the pandemic on routine variables. The variables selected for the study were tested with Chi-Square test. Variables with *p* < 0.20 were included in the binary logistic regression model according to the step forward method. In the present study, the exponent of β was used as OR, the measure of association. A level of significance of 5% was adopted.

The present study was conducted in accordance with the ethical norms and guidelines of Resolution No. 466/2012 of the National Health Council, Ministry of Health, and submitted to the Research Ethics Committee of the School of Medicine of Jundiaí. The municipality’s Health Management and Promotion Unit and Primary Care Department were contacted to request authorization for data collection. Data collection was only started after approval of the study by the Ethics Committee (Ethical Clearance Certificate: 31748920.1.0000.5412). The questionnaire was applied just after the explanation of the study objectives and after the participants had signed the free informed consent form. For illiterate participants, consent to participate was obtained from their legal guardians (there was a consent form for legal guardians).

## 3. Results

The total sample was 2432 respondents who answered the questionnaire. Most respondents were female, between 18 and 59 years old, and self-reported white; participants interviewed at the UBS had an educational attainment up to high school and those interviewed in the household had higher education or a postgraduate course. In addition, most respondents had an income of 5 MW or more and there were no health professionals in the majority of households. In the UBS, we received a response from 1181 individuals with respiratory symptoms who sought health units for consultation and rapid testing. In this group, there was a predominance of women (63.7%), age between 18 and 39 years (52.8%), and self-reported white skin color (67.2%) (Table 1). In the second stage, the data collected in the households comprised 1251 individuals, with a predominance of women (64.3%), age between 40 and 59 years (39.7%), and self-reported white skin color (72.3%) (Table 1).

A large number of respondents were concerned about losing someone in their family (86.0%) and reported fear of becoming infected (69.8%) or that a family member would become infected (85.7%) (Table 2). The majority of the population worried a lot about school closure (65.0%) (Table 2).

Regarding socioeconomic variables, the most concerned age groups were individuals aged 18 to 39 (OR: 2.89; CI: 2.30–3.63; *p* < 0.001) and 40 to 59 years (OR: 2.16; CI: 1.72–2.72; *p* < 0.001), as well as females (OR: 1.60; CI: 1.35–1.89; *p* < 0.001), the lower income population (up to 1 MW [OR: 2.49; CI: 1.78–3.48; *p* < 0.001] and 1 to 3 MW [OR: 1.73; CI: 1.45–2.06; *p* < 0.001]), black or mixed race participants (OR: 1.67; CI: 1.40–1.99; *p* < 0.001), and respondents who had completed primary or secondary education ([OR: 1.376; CI: 1.09–1.73; *p* = 0.006] and [OR: 1.660; CI: 1.39–1.909; *p* < 0.001], respectively), excluding illiterate individuals (Table 3).

With respect to behavioral variables, we observed that respondents who only left home for work were more concerned (OR: 1.99; CI: 1.21–3.26; *p* = 0.006), as were those who adhered to the isolation guidelines (OR: 1.43; CI: 1.20–1.71; *p* < 0.001), who no longer had an income or whose income had decreased ([OR: 3.34; CI: 1.89–5.87; *p* < 0.001] and [OR: 1.92; CI: 1.62–2.26; *p* < 0.001], respectively), and those who became unemployed (OR: 2.24; CI: 1.72–2.94; *p* < 0.001). Respondents who exhibited any symptoms were also more concerned (OR: 1.40; CI: 1.14–1.73; *p* = 0.001) (Table 3).

Logistic regression model showed that women (OR: 1.42; CI: 1.18–1.71; *p* = 0.003) were 1.42 times more likely than man to present more prevalence of concern above the median. Black and mixed people presented 1.40 times more concern prevalence than whites (OR: 1.40; CI: 1.15–1.71; *p* < 0.01), and those with less than 3 minimum wage were more likely to have more concern than those with 3 MW or more (OR: 2.58; IC: 1.80–3.70; *p* < 0.01 and OR: 1.64; IC: 1.35–1.98; *p* < 0.01. Also, younger than 60 years old presented more prevalence than the older (Table 4).

## 4. Discussion

The present data obtained for a population sample using an instrument proposed by the WHO showed that the COVID-19 pandemic was associated with an intense perception of concern, especially among females, black individuals, individuals with a lower income, and individuals aged between 18 and 59 years. The aspects involved with concern during the COVID-19 pandemic included: fear of losing someone, fear of catching coronavirus and contaminating family or someone close, and socioeconomic aspects. A study carried out in Latin America and Spain found similar results regarding mental health and self-perceived health [22]. In this study, the prevalence of anxiety and depression decreased as age increased, socioeconomic condition, and being women were associated with worse mental health and poor self-perceived health [22].

The final regression model showed that women were more affected. Women are perceived as more aware of their health and more willing to receive health information [13], probably because they more frequently seek health services and are more concerned about the health of their relatives. Women had a greater perception of concern, as also observed in studies conducted in Brazil [13], Saudi Arabia [23] and China [24] and in a systematic review assessing the general population [25]. A literature review observed that among women studies in pandemic, anxiety, depression, and stress were the common outcomes. Socioeconomic aspects such as lower education and income, preexisting mental health problems, and living alone or with children were risk factors for higher levels of anxiety and depression among women [26].

Black or mixed-race individuals had a greater perception of concern in this study. This finding may be due to a multiplicity of factors historically linked to the black population such as poor social determinants of health [27]; higher rates of unemployment and illiteracy may have influenced the outcome analyzed since this population would start the evaluation with poorer indices, demonstrating this health inequity and highlighting the importance of actions to correct this avoidable history.

Two household income groups had worse levels of concern: up to 1 MW and up to 3 MW. An income of up to 1 MW has been recognized as a risk factor for poor health perception [28]. A systematic review [27] comparing the mental health situation before and after declaration of the pandemic identified higher income as a protective factor against emotional stress. The same result was reported by an epidemiological survey [29] conducted in São Paulo in 2021 and as found in Latin America and Spain [22].

Age was also a factor related to greater perception of concern. Although mortality has been shown to be higher in the older population [4], an epidemiological study [19] found more positive cases in the younger and economically active population, as did the nationwide study conducted by the University of Pelotas, called EPICOVID19 [30]. It is possible that being part of the population most affected by a previously unknown disease, whose mortality profile had not yet been defined, contributed to the greater perception of concern among these age groups—even though the highest mortality was not observed at these ages. Study found that younger people presented poorer self-perceived health and mental health [22].

Regarding the answers to questions about the main aspects of concern, the fear of losing a loved one was the most frequent worry, followed by the fear of a family member contracting the disease, which was more frequent among symptomatic individuals as it was found in study carried out in EUA [7] where almost half (48%) of Americans felt anxious about the possibility of contracting the new coronavirus and 40% were anxious about the possibility of developing serious illness or dying from the virus. These results are consistent with those found in Brazil [13] using the other scales. It is understandable that the loss of a family member was a concern considering the alarming increase in the number of reported deaths [4] each day throughout all media.

The initial hypothesis was that change in routine, social isolation, and having the disease would contribute to worsening emotional stress and in fact they represent factors individually associated in univariate analysis. However, the logistic regression analysis did not confirm this hypothesis, demonstrating greater importance of other socioeconomic and demographic factors for the condition studied. The same was observed for the presence or absence of symptoms, the situation of family members, and the COVID-19 test result.

One limitation of the study was the use of data from a seroepidemiological study [19] in which the sample size calculation and method of sample selection were defined for the outcome “COVID-19 rapid test result”; thus, representativeness of the population cannot be inferred for the outcome “perception of concern”. The difficulty in training the data collection teams during the critical phase of the pandemic was overcome with the use of an application and virtual training. A non-validated questionnaire was used to measure concern since the study was conducted at the beginning of the pandemic when no validated instruments were available to assess the concern caused by the pandemic; however, the Behavioural Insights on COVID-19 tool recommended by the WHO was adopted. Instruments to measure the concern in health crises or disasters should be validated, considering good psychometrics properties, that could improve the scale used. The Cronbach’s alpha was 0.67 and the adequate value was 0.70 (very close) [21]. It is important considering the possibility of information bias on this study, that may interfere with the results.

The present study addresses a relevant issue using important sociodemographic characterization, as many COVID-19 studies were conducted in outpatient settings or used public data from notification forms. The data obtained in this population-based study using an instrument proposed by the WHO indicate great perceived concern caused by the pandemic, as demonstrated by the sociodemographic and behavioral factors associated with this condition. The city of Jundiaí is representative of other similar cities in Brazil and abroad and the large, randomized sample allowed a representation of the city’s population.

It should be highlighted that the intense economic instability during the pandemic and the lack of public policies to help vulnerable families since the beginning of the pandemic may have influenced the results of the present study since these factors were associated with a greater perception of concern. Social security policies in these public health emergencies can be important to reduce worries, which are already intense due to the large number of reported daily deaths. Likewise, programs that reduce racial and gender inequities are necessary in the pandemic context. The importance of early observation of psychosocial risk factors such as concern, which may predispose to more serious mental illnesses, is also highlighted. Disorders such as anxiety and depression have been identified as post-COVID-19 sequelae, especially among hospitalized patients [31].

The collected data raises awareness of public health providers towards the importance of specific programs that provide care to vulnerable populations during health crisis that may occur at any given time such as pandemics, natural disasters or climatic changes. These populations have specific needs that must be addressed in order to lessen the impact of such events. It was found that during pandemic the risk perception was high, and the emotional response was strong in a study carried out in Bosnia and Herzegovina [32]. The same study evaluated the perception of trust in public polices and institutions that were low emphasizing the need to improve health literacy to help reinforce protective behaviors in public health [32].

## 5. Conclusions

Our study demonstrates that the COVID-19 pandemic had a negative impact on the mental health of the population studied, causing an intense perception of concern related to social determinants of health, with a greater impact on the female population, individuals aged 18 to 59, individuals earning up to 3 MW, and self-reported Blacks. Although important, behavioral factors or the presence of symptomatic individuals in the household were not determinants of greater concern in the population studied. We highlight the importance of this survey for the preparation of health services, suggesting the training of teams in the identification of sociodemographic factors that indicate an increased concern regarding the consequences of the pandemic for the mental health of the population. This preparation is expected to prevent the progression of concern to more severe mental disorders. Also, policies that reduce socioeconomic, race and gender inequalities must be developed and taken in consideration. Future studies should be carried out monitoring over time mental health and aspects that impact on the population in health crises.

## Figures and Tables

**Figure 1 ijerph-22-01293-f001:**
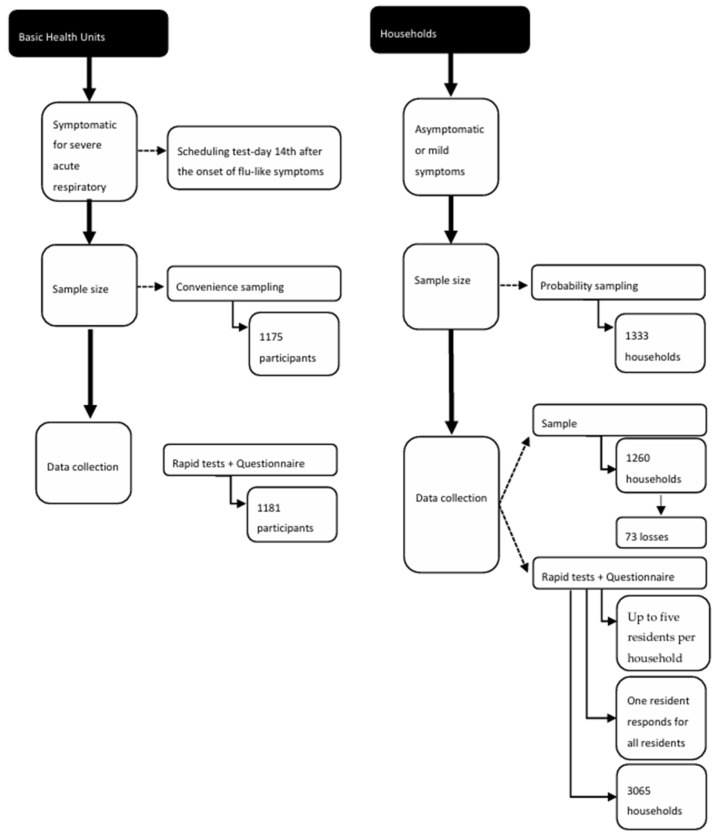
Data collection procedure of the studies conducted at the Basic Health Units and households, Jundiaí, São Paulo, 2020.

**Table 1 ijerph-22-01293-t001:** Sociodemographic characteristics of the participants who answered the questionnaires in the households and UBS. Jundiaí, SP, 2020.

Variable		Household		UBS		Total	
		*n*	%	*n*	%	*n*	%
Sex	Female	804	64.3	752	63.7	1556	64.0
	Male	447	35.7	429	36.3	876	36.0
Age (years)	18 to 39	333	26.6	624	52.8	957	39.4
	40 to 59	497	39.7	441	37.3	638	26.2
	≥60	414	33.1	116	9.8	530	21.8
	NR	4	0.3	0	0.0	4	0.2
Skin color	Black	69	5.5	73	6.2	82	3.4
	Mixed race	244	19.5	296	25.1	340	14.0
	White	905	72.3	794	67.2	1699	69.9
	Yellow	30	2.4	10	0.8	40	1.6
	NR	3	0.2	8	0.7	11	0.5
Educational level	Illiterate	16	1.3	0	0.0	16	0.7
	Up to elementary school	292	23.3	146	12.4	438	18.0
	Up to high school	446	35.7	562	47.6	1008	41.4
	Complete higher education/Postgraduation	496	39.6	458	38.8	954	39.2
	NR	1	0.1	15	1.3	16	0.7
Household income	<1 MW	82	6.6	56	4.7	138	5.7
	1 to 2 MW	197	15.7	214	18.1	411	16.9
	2 to 3 MW	241	19.3	218	18.5	432	17.8
	3 to 4 MW	227	18.1	243	20.6	270	11.1
	≥5 MW	355	28.4	337	28.5	692	28.5
	No income	11	0.9	13	1.1	24	1.0
	NR	138	11.0	100	8.5	238	9.8
Highest educational attainment of the household	Illiterate	7	0.6	0	0.0	7	0.3
	Up to elementary school	85	6.8	28	2.4	113	4.6
	Up to high school	358	28.6	380	32.2	738	30.3
	Complete higher education/Postgraduation	783	62.6	727	61.6	1510	62.1
	NR	18	1.4	46	3.9	64	2.6
Health professional in the household	Yes	225	18.0	284	24.0	509	20.9
	No	1026	82.0	897	76.0	1923	79.1
Total		1251	100	1181	100	2432	100

MW: minimum wage (1 MW = R$ 1045.00); NR: not reported.

**Table 2 ijerph-22-01293-t002:** Perception of concern of individuals regarding the COVID-19 pandemic. Jundiaí, SP, 2020.

		Household		UBS		Total	
		*n*	%	*n*	%	*n*	%
Fear of being very alone	Does not worry me	747	59.7	538	45.6	1285	52.8
	Worries me a little	262	20.9	233	19.7	495	20.4
	Worries me a lot	242	19.3	410	34.7	652	26.8
Fear of losing someone close due to the disease	Does not worry me	134	10.7	53	4.5	187	7.7
	Worries me a little	110	8.8	43	3.6	153	6.3
	Worries me a lot	1007	80.5	1085	91.9	2092	86.0
Fear of a family member/friend contracting coronavirus	Does not worry me	93	7.4	60	5.1	153	6.3
	Worries me a little	125	10.0	70	5.9	195	8.0
	Worries me a lot	1033	82.6	1051	89.0	2084	85.7
Experiencing financial difficulties	Does not worry me	632	50.5	495	41.9	1127	46.3
	Worries me a little	243	19.4	202	17.1	445	18.3
	Worries me a lot	376	30.1	484	41.0	860	35.4
School closure	Does not worry me	221	17.7	226	19.1	447	18.4
	Worries me a little	209	16.7	194	16.4	403	16.6
	Worries me a lot	821	65.6	761	64.4	1582	65.0
Fear of catching coronavirus	Does not worry me	167	13.3	254	21.5	421	17.3
	Worries me a little	173	13.8	140	11.9	313	12.9
	Worries me a lot	911	72.8	787	66.6	1698	69.8
Fear of becoming unemployed	Does not worry me	571	45.6	330	27.9	901	37.0
	Worries me a little	108	8.6	107	9.1	215	8.8
	Worries me a lot	572	45.7	744	63.0	1316	54.1
Fear of dying	Does not worry me	376	30.1	372	31.5	748	30.8
	Worries me a little	162	12.9	136	11.5	298	12.3
	Worries me a lot	713	57.0	673	57.0	1386	57.0
Fear of shortage of food and basic items caused by the pandemic	Does not worry me	508	40.6	431	36.5	939	38.6
	Worries me a little	240	19.2	215	18.2	455	18.7
	Worries me a lot	503	40.2	535	45.3	1038	42.7

**Table 3 ijerph-22-01293-t003:** Perception of concern and associated variables among individuals who answered the questionnaire. Jundiaí, SP, 2020.

		Concern							
		Below the Median (≤22)		Above the Median (>22)					
		*n*	%	*n*	%	Unadjusted OR	95% CI		*p*
Age (years)	18 to 39	445	45.5	512	53.5	2.89	2.30	3.63	<0.001
	40 to 59	504	53.7	434	46.3	2.16	1.72	2.72	<0.001
	≥60	379	51.5	151	28.5	1			
Sex	Female	788	50.7	767	49.3	1.60	1.35	1.89	<0.001
	Male	544	62.1	332	37.9	1			
Income (minimum wage)	<1	712	61.3	450	38.7	2.49	1.78	3.48	<0.001
	1 to 3	416	47.8	454	52.2	1.73	1.45	2.06	<0.001
	≥3	63	38.9	99	61.1	1			
Skin color	Black and mixed race	309	45.3	373	54.7	1.67	1.40	1.99	<0.001
	White	986	58.1	712	41.9	1			
Educational level	Illiterate	15	93.8	1	6.2	1	0.45	3.35	0.67
	Up to elementary school	336	81.8	75	18.2	1.37	1.09	1.73	0.006
	Up to high school	713	75.1	237	24.9	1.66	1.39	1.99	<0.001
	Complete higher education/Postgraduation	744	81.8	165	18.2	1			
Did your routine change?	Not going out or going to the market, pharmacy, emergency	875	57.7	642	42.3	1.38	0.85	2.25	0.19
	Going out to work	408	48.6	431	51.4	1.99	1.21	3.26	0.006
	No change in routine	49	65.3	26	34.7	1			
Did you adhere to prevention guidelines?	Yes	1128	54.9	925	45.1	0.96	0.77	1.20	0.72
	No	204	54.0	174	46.0	1			
Did you adhere to isolation?	Yes	991	57.3	737	42.7	1.43	1.20	1.71	<0.001
	No	340	48.4	362	51.6	1.00			
Was your income affected?	No longer having income	19	33.9	37	66.1	3.34	1.89	5.87	<0.001
	Decreased a little	556	47.2	622	52.8	1.92	1.62	2.26	<0.001
	Continues the same	721	63.1	421	36.9	1			
Were you unemployed?	Yes	94	37	160	63.0	2.24	1.716	2.935	<0.001
	No	1238	56.9	939	43.1	1			
Did you have any symptom?	Yes	210	47.8	229	52.2	1.40	1.14	1.73	0.001
	No	1122	56.3	870	43.7	1			
Did a family member have any symptom?	Yes	173	49.4	11	50.6	1.28	1.02	1.61	0.03
	No	1159	55.7	922	44.3	1			
Did any family member test positive for COVID-19?	Yes	1106	55.8	875	44.2	1.25	1.02	1.54	0.03
	No	226	50.2	224	49.8	1			
Did you lose a family member?	Yes	73	58.9	51	41.1	1.19	0.82	1.72	0.35
	No	1259	545.6	45.4		1			
COVID-19 test	Positive	1120	55.7	889	44.3	1.25	1.01	1.54	0.04
	Negative	212	50.2	210	49.8	1			

**Table 4 ijerph-22-01293-t004:** Logistic regression model of factors associated with the perception of concern. Jundiaí, SP, 2020.

Variable		Adjusted OR	95% CI	*p*
Sex	Female	1.42	1.18–1.71	0.003
	Male	1		
Skin color	Black and mixed race	1.40	1.15–1.71	<0.01
	White	1		
Income (minimum wage)	<1	2.58	1.80–3.70	<0.01
	1 to 3	1.64	1.35–1.98	<0.01
	≥3	1		
Age (years)	18 to 39	3.07	2.39–3.94	<0.01
	40 to 59	2.42	1.89–3.10	<0.01
	≥60	1		

## Data Availability

The datasets used can be accessed on OSF Home: https://osf.io/amsxn/?view_only=46449a9acdfc44219551b5cfd56d8a62 (accessed on 20 April 2023).

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
