# Peer review of "Perception of Concern and Associated Factors During the COVID-19 Pandemic: An Epidemiological Survey in a Brazilian Municipality"

_ijerph, 2025, doi:10.3390/ijerph22081293_

Round 1
Reviewer 1 Report
Comments and Suggestions for Authors
Thank you for allowing me to review the paper titled: “Perception of concern and associated factors during the COVID-19 pandemic: an epidemiological survey in a Brazilian municipality”
Abstract
Include a brief mention of the importance of these findings for public mental health policies or crisis interventions.
Introduction
More clearly highlight the gap in the literature and the specific value of this study within the Brazilian context.
Methodology
Detail how the random selection of households was carried out—specifically, which neighborhoods were selected and how this process ensured geographic representation of the population.
Provide further information on how the WHO questionnaire was adapted, including a description of how its questions were adjusted to reflect Brazilian culture and context.
Explain how the binary logistic regression model was constructed, and justify the inclusion of variables based on the proposed significance level in the bivariate analysis.
Results
Present the most relevant findings in the text before introducing the tables.
Clarify how socioeconomic factors—such as sex, race, and income—influence the perception of concern in the regression analysis.
Explain the statistical values (e.g., p-values and odds ratios) in the context of the study’s findings, making their practical implications clearer.
Discussion
Relate the findings to previous studies, particularly those conducted in Brazil or in similar social contexts.
Expand on the practical implications of the results, especially in terms of public policies related to mental health.
Further elaborate on the study’s limitations and discuss how these may have affected the results.
Conclusions
Propose concrete recommendations for mental health interventions in future crises similar to the COVID-19 pandemic.
Highlight and suggest specific actions to reduce racial and socioeconomic inequities, emphasizing the contribution of this study to the development of public health policies.
Recommend future research directions, particularly longitudinal studies that can explore these associations over time.
Author Response
Reviewer 1
Abstract
Include a brief mention of the importance of these findings for public mental health policies or crisis interventions.
Reply: This information was included in the abstract.
Introduction
More clearly highlight the gap in the literature and the specific value of this study within the Brazilian context.
Reply: We have included the information at the end of the paragraph that begins with the sentence “To understand the psychosocial”.
Methodology
Detail how the random selection of households was carried out—specifically, which neighborhoods were selected and how this process ensured geographic representation of the population.
Reply: The details of the selection of households are well described in a cited manuscript, includig a map of the neighborhoods. To avoid plagiarism and take advantage of the space in the manuscript to bring in something new, such as the WHO questionnaire and the concern, we only cited the other article. All the neighborhoods were selected and the number of participants was proportional to the number of inhabitants in the neighborhood so that the sample was representative of the municipality. The draw was made from an address list of all the homes registered with the municipality's tax (Batista MJ, Lino CM, Tenani CF, Zanin L, Correia da Silva AT, Nunes Lipay MV. Seroepidemiological investigation of COVID-19: A cross-sectional study in Jundiai, São Paulo, Brazil. Clapham HE, editor. Glob. Public Health. 2022 Sep 9 [cited 2023 Jan 13];2(9): e0000460. Available from: https://journals.plos.org/globalpublichealth/article?id=10.1371/journal.pgph.0000460).
Provide further information on how the WHO questionnaire was adapted, including a description of how its questions were adjusted to reflect Brazilian culture and context.
Reply: In the context of the pandemic, there was no time to validate the questionnaire. It was therefore translated, adapted and tested by experts and during staff training, before being used for data collection. We added more details about the WHO questionnaire and how was selected the concern questions.
Explain how the binary logistic regression model was constructed, and justify the inclusion of variables based on the proposed significance level in the bivariate analysis.
Reply: More information has been added on the logistic regression analysis and also on the selection of variables to enter the model. The p-value selection criterion was for inclusion in the regression model, considering that the choice of variables for the study was based on the literature and the questionnaire proposed by the WHO.
Results
Present the most relevant findings in the text before introducing the tables.
Reply: The most relevant findings in the text were presented before introducing the tables.
Clarify how socioeconomic factors—such as sex, race, and income—influence the perception of concern in the regression analysis.
Reply: These aspects were included in the discussion.
Explain the statistical values (e.g., p-values and odds ratios) in the context of the study’s findings, making their practical implications clearer.
Reply: The statistical values were better explained.
Discussion
Relate the findings to previous studies, particularly those conducted in Brazil or in similar social contexts.
Reply: The discussion was modifed as the suggestion above.
Expand on the practical implications of the results, especially in terms of public policies related to mental health.
Reply: The practical implications was added.
Further elaborate on the study’s limitations and discuss how these may have affected the results.
Reply: We added this infromation at the limitations topic.
Conclusions
Propose concrete recommendations for mental health interventions in future crises similar to the COVID-19 pandemic.
Reply: We added this information.
Highlight and suggest specific actions to reduce racial and socioeconomic inequities, emphasizing the contribution of this study to the development of public health policies.
Reply: Thank you for the suggestions we added this information.
Recommend future research directions, particularly longitudinal studies that can explore these associations over time.
Reply: We added this information.
Reviewer 2 Report
Comments and Suggestions for Authors
Thanks for the opportunity to review this study. Please see my comments below.
Introduction
-
It feels somewhat abrupt to begin the introduction with a definition of psychosocial, especially since the term is not mentioned in the first or second paragraphs. I recommend moving this sentence to a later point—ideally just before it is directly introduced or discussed in the context of the study.
-
The third paragraph lacks clarity in terms of its relevance to the current study. While it appears to summarize literature on psychosocial effects, the connection to the present research is unclear. If these studies are meant to highlight existing gaps, those gaps should be explicitly stated. Otherwise, consider streamlining this section to better align with the research rationale.
-
The statement:
“Isolation can cause depression and anxiety, especially in older people; the closure of schools and the need to teach children at home can increase stress and cases of violence.” is currently missing a citation. Please provide a supporting source. -
I would like to see more elaboration on the statement:
“The psychosocial changes of COVID-19 are therefore multifactorial and are observed in multiple layers.” Expanding this with specific examples or referring to a conceptual framework would strengthen the argument. -
The following sentence requires a citation:
“Furthermore, a peculiar syndrome called ‘headline stress disorder’ has been observed during modern pandemics.”
Overall, the introduction lacks organizational clarity. To improve readability and coherence, I suggest using subheadings to structure the background more clearly (e.g., “Psychosocial Impacts of COVID-19,” “Gaps in the Literature,” etc.).
Method
It would improve clarity to present each outcome variable in a separate subsection. For each variable, please include: A clear definition, Example item(s) from the scale, Evidence of the scale’s reliability and validity (e.g., Cronbach’s alpha)
This level of detail would enhance the transparency and replicability of the study.
Discussion
The discussion currently lacks depth and novelty. Many of the findings appear to replicate established results, which is valuable but not sufficiently highlighted in the current framing. To enhance the impact of the discussion: Clearly articulate how these findings align with or differ from existing studies. Emphasize any unique or unexpected results. Discuss the broader implications of the findings and any contributions to theory, practice, or policy.
Author Response
Introduction
- It feels somewhat abrupt to begin the introduction with a definition of psychosocial, especially since the term is not mentioned in the first or second paragraphs. I recommend moving this sentence to a later point—ideally just before it is directly introduced or discussed in the context of the study.
Reply: Thank you for the suggestion. We moved this information in introduction section to the end of the paragraph.
- The third paragraph lacks clarity in terms of its relevance to the current study. While it appears to summarize literature on psychosocial effects, the connection to the present research is unclear. If these studies are meant to highlight existing gaps, those gaps should be explicitly stated. Otherwise, consider streamlining this section to better align with the research rationale.
Reply: The third paragraph bring results from questions very similar of our study, as fear of losing someone close due to the disease, fear of a family member or friend contracting coronavirus, fear of going through financial difficulties, school closure, fear of catching coronavírus. However we modified the frase in order to make sense in the introduction.
- The statement:
“Isolation can cause depression and anxiety, especially in older people; the closure of schools and the need to teach children at home can increase stress and cases of violence.” is currently missing a citation. Please provide a supporting source.
Reply: We provided that reference and corrected the meaning of the frase. We also added on more reference.
- I would like to see more elaboration on the statement:
“The psychosocial changes of COVID-19 are therefore multifactorial and are observed in multiple layers.” Expanding this with specific examples or referring to a conceptual framework would strengthen the argument.
Reply: We thank you for the comment. We added information explaining better the statement.
- The following sentence requires a citation:
“Furthermore, a peculiar syndrome called ‘headline stress disorder’ has been observed during modern pandemics.”
Reply: We removed this sentence in order to better align with the research rationale.
Overall, the introduction lacks organizational clarity. To improve readability and coherence, I suggest using subheadings to structure the background more clearly (e.g., “Psychosocial Impacts of COVID-19,” “Gaps in the Literature,” etc.).
Reply: We thank you very much the suggestion and organize the introduction as sugested but without the subheadings considering the short size of the introduction in the manuscript.
Psychosocial Impacts of COVID-19- paragraphs 1 to 7
Concern - paragraph 8
Gaps in the literature – paragraph 8
Behavioural Insights on COVID-19- paragraph 9
Justification- paragraph 10
Method
It would improve clarity to present each outcome variable in a separate subsection. For each variable, please include: A clear definition, Example item(s) from the scale, Evidence of the scale’s reliability and validity (e.g., Cronbach’s alpha).
This level of detail would enhance the transparency and replicability of the study.
Reply: We added details about the scale applied, but there was no evidence about the reliability and validity of the scale, because as we discussed the Behavioral Insights was created during the pandemic and sugested by WHO to be applied in different contexts. In the documment of the questionnaire we can find the recommendations to adapt for each country. It was a very important tool during the pandemic.
We found other studies that used the same questionnaire. We carried out the Cronbach’s alpha that was 0,67, considered a moderate internal consistency.
Discussion
The discussion currently lacks depth and novelty. Many of the findings appear to replicate established results, which is valuable but not sufficiently highlighted in the current framing. To enhance the impact of the discussion: Clearly articulate how these findings align with or differ from existing studies. Emphasize any unique or unexpected results. Discuss the broader implications of the findings and any contributions to theory, practice, or policy.
Reply: We thank you for the suggestion. The discussion was reformulated to attend the reviewers comments.